# Correlation between Histopathological Prognostic Tumor Characteristics and [^18^F]FDG Uptake in Corresponding Metastases in Newly Diagnosed Metastatic Breast Cancer

**DOI:** 10.3390/diagnostics14040416

**Published:** 2024-02-14

**Authors:** Jorianne Boers, Bertha Eisses, Mieke C. Zwager, Jasper J. L. van Geel, Frederike Bensch, Erik F. J. de Vries, Geke A. P. Hospers, Andor W. J. M. Glaudemans, Adrienne H. Brouwers, Martijn A. M. den Dekker, Sjoerd G. Elias, Evelien J. M. Kuip, Carla M. L. van Herpen, Agnes Jager, Astrid A. M. van der Veldt, Daniela E. Oprea-Lager, Elisabeth G. E. de Vries, Bert van der Vegt, Willemien C. Menke-van der Houven van Oordt, Carolina P. Schröder

**Affiliations:** 1Department of Medical Oncology, University Medical Center Groningen, University of Groningen, 9713 Groningen, The Netherlands; j.boers@umcg.nl (J.B.); b.eisses@umcg.nl (B.E.); j.j.l.van.geel@umcg.nl (J.J.L.v.G.); f.bensch@umcg.nl (F.B.); g.a.p.hospers@umcg.nl (G.A.P.H.); e.g.e.de.vries@umcg.nl (E.G.E.d.V.); 2Department of Pathology, University Medical Center Groningen, University of Groningen, 9713 Groningen, The Netherlands; m.c.zwager@umcg.nl (M.C.Z.); b.van.der.vegt@umcg.nl (B.v.d.V.); 3Department of Nuclear Medicine and Molecular Imaging, University Medical Center Groningen, University of Groningen, 9713 Groningen, The Netherlands; e.f.j.de.vries@umcg.nl (E.F.J.d.V.); a.w.j.m.glaudemans@umcg.nl (A.W.J.M.G.); a.h.brouwers@umcg.nl (A.H.B.); 4Department of Radiology, University Medical Center Groningen, University of Groningen, 9713 Groningen, The Netherlands; m.ddekker@ziggo.nl; 5Department of Epidemiology, Julius Center for Health Sciences and Primary Care, University Medical Center Utrecht, Utrecht University, 3584 Utrecht, The Netherlands; s.g.elias@umcutrecht.nl; 6Department of Medical Oncology, Radboud Medical Center, 6500 Nijmegen, The Netherlands; evelien.kuip@radboudumc.nl (E.J.M.K.); carla.vanherpen@radboudumc.nl (C.M.L.v.H.); 7Department of Medical Oncology, Erasmus MC Cancer Institute, 3015 Rotterdam, The Netherlands; a.jager@erasmusmc.nl (A.J.); a.vanderveldt@erasmusmc.nl (A.A.M.v.d.V.); 8Department of Radiology and Nuclear Medicine, Amsterdam University Medical Center, Location VU University Medical Center, 1081 Amsterdam, The Netherlands; d.oprea-lager@amsterdamumc.nl; 9Department of Medical Oncology, Amsterdam University Medical Center, Location VU University Medical Center, 1081 Amsterdam, The Netherlands; c.menke@amsterdamumc.nl; 10Department of Medical Oncology, Dutch Cancer Institute, 1066 Amsterdam, The Netherlands

**Keywords:** [^18^F]FDG-PET, breast cancer, metastases, histopathological characteristics

## Abstract

Background: In metastatic breast cancer (MBC), [^18^F]fluorodeoxyglucose positron emission tomography/computed tomography ([^18^F]FDG-PET/CT) can be used for staging. We evaluated the correlation between BC histopathological characteristics and [^18^F]FDG uptake in corresponding metastases. Patients and Methods: Patients with non-rapidly progressive MBC of all subtypes prospectively underwent a baseline histological metastasis biopsy and [^18^F]FDG-PET. Biopsies were assessed for estrogen, progesterone, and human epidermal growth factor receptor 2 (ER, PR, HER2); Ki-67; and histological subtype. [^18^F]FDG uptake was expressed as maximum standardized uptake value (SUV_max_) and results were expressed as geometric means. Results: Of 200 patients, 188 had evaluable metastasis biopsies, and 182 of these contained tumor. HER2 positivity and Ki-67 ≥ 20% were correlated with higher [^18^F]FDG uptake (estimated geometric mean SUV_max_ 10.0 and 8.8, respectively; *p* = 0.0064 and *p* = 0.014). [^18^F]FDG uptake was lowest in ER-positive/HER2-negative BC and highest in HER2-positive BC (geometric mean SUV_max_ 6.8 and 10.0, respectively; *p* = 0.0058). Although [^18^F]FDG uptake was lower in invasive lobular carcinoma (*n* = 31) than invasive carcinoma NST (*n* = 146) (estimated geometric mean SUV_max_ 5.8 versus 7.8; *p* = 0.014), the metastasis detection rate was similar. Conclusions: [^18^F]FDG-PET is a powerful tool to detect metastases, including invasive lobular carcinoma. Although BC histopathological characteristics are related to [^18^F]FDG uptake, [^18^F]FDG-PET and biopsy remain complementary in MBC staging (NCT01957332).

## 1. Introduction

Breast cancer is the most common type of cancer diagnosed in women [1]. For the workup of metastatic breast cancer, [^18^F]fluorodeoxyglucose positron emission tomography accompanied with computed tomography ([^18^F]FDG-PET/CT) can be used according to the European Society of Medical Oncology (ESMO) guidelines [2]. Regarding bone metastases, [^18^F]FDG-PET compared to conventional bone scintigraphy leads to clinically relevant differences in metastatic breast cancer management in 16% of patients [3]. [^18^F]FDG-PET can detect more bone lesions than conventional bone scintigraphy [3]. Aside from information on the staging of breast cancer, [^18^F]FDG-PET also yields information on metabolic activity [4,5]. Cancer cells are typically more metabolically active and thus show higher glucose uptake. [^18^F]FDG, a glucose analog, is transported by membrane-specific glucose transporters (GLUT) into the cell and is phosphorylated by hexokinase to [^18^F]FDG 6-phosphate. Since phosphorylation of [^18^F]FDG -6-phosphate cannot take place, it is trapped in the cell [6]. Tumor glucose use can be quantified by the PET-derived parameter maximum standardized uptake value (SUV).

Routinely, a metastasis biopsy is performed for the detection of tumors and to assess the presence of human epidermal growth factor receptor 2 (HER2) and histological subtype. [^18^F]FDG-PET has been investigated to assess the presence of tumor and known prognostic histopathological characteristics, mostly of the primary tumor. Hormone receptor status and Ki-67 proliferation index have been described to affect the glucose metabolism of tumors, resulting in differences in [18F]FDG-PET SUVs [7,8]. For example, ER-negative lesions are most commonly associated/correlated with higher uptake, suggesting accelerated glucose metabolism. This suggests more glucose metabolism is needed to meet the energy demand for rapid growth [9]. Microarray analysis confirms these data and identifies genes associated with increased glucose use as measured by PET [10]. Ki67 was also found to be strongly correlated with SUV [11,12]. The use of [^18^F]FDG-PET was discouraged in invasive lobular carcinoma, due to lower [^18^F]FDG uptake in invasive lobular carcinoma compared to invasive carcinoma NST in the primary setting [13]. Whether this is also the case in the metastatic setting is not clear.

At present, due to varying technical (scan) aspects and low-quality evidence, such as no head-to-head comparison between [^18^F]FDG-PET and biopsy as the gold standard, thus far it remains unclear whether [^18^F]FDG uptake is related to histopathological characteristics in metastatic breast cancer [14]. Therefore, we assessed in patients participating in the IMPACT metastatic breast cancer study the correlation between histopathological characteristics of breast cancer and [^18^F]FDG uptake in corresponding metastases.

## 2. Methods

### 2.1. Patients and Study Design

Patients with first presentation of non-rapidly progressive (defined as not requiring urgent initiation of chemotherapy) metastatic breast cancer, regardless of their breast cancer subtype, were enrolled in the multicenter IMPACT metastatic breast cancer study (NCT01957332, the IMPACT breast trial (2013/146), was approved by the Medical Ethics Committee of the UMCG and supported by the Dutch Cancer Society [Grant 2012-5565]) between August 2013 and May 2018. The study was performed at four Dutch university centers: University Medical Center Groningen (UMCG), Amsterdam University Medical Center (Amsterdam UMC location VUmc), Radboud University Medical Center (Radboud UMC), and Erasmus Medical Center (Erasmus MC). It was approved by the Medical Ethical Committee of the UMCG and the Central Committee on Research Involving Human Subjects. All patients provided written informed consent. The primary aim of the IMPACT metastatic breast cancer study was to assess the clinical utility of molecular imaging, in addition to standard diagnostics. Patients underwent standard diagnostic assessments at baseline, including [^18^F]FDG-PET/CT and a core needle biopsy of the metastasis, before treatment initiation. The detailed methods for the IMPACT study have been published previously [3,15]. For the present sub-study, baseline [^18^F]FDG-PET, contrast-enhanced CT (ceCT), and histopathological characteristics were used for all patients of whom a metastasis could be biopsied. Target lesions were defined according RECIST 1.1.

As the outcome and therapy response data are currently still being analyzed, the baseline data are presented here. 

### 2.2. Metastasis Biopsy and Histopathological Characteristics

A core needle biopsy with 2–3 samples (of approximately 22 mm length) from a suspected metastasis was obtained before or after [^18^F]FDG-PET, but before starting treatment according to standard clinical care procedures. The biopsy site was determined by conventional imaging and the biopsy procedure was guided by CT or ultrasound, with the exception of skin lesions, which were visually identified and biopsied. Bone biopsies were allowed as decalcification with EDTA does not affect hormone and HER2 receptor status [15]. Biopsies were formalin-fixed and paraffin-embedded and centrally reviewed by an experienced breast pathologist (BvdV) blinded for the imaging results. Immunohistochemistry (IHC) for estrogen receptor (ER) (SP-1, Ventana/Roche, Illkrich, France), progesterone receptor (PR) (1E2, Ventana/Roche), and HER2 (SP3, Thermo Fisher Scientific, Waltham, Massachusetts, USA) was performed on an automated staining platform (Ventana Benchmark Ultra Ventana/Roche) according to the manufacturer’s protocols. All antibodies were pre-diluted by the supplier. According to Dutch guidelines, ER and PR status were considered positive if ≥10% of the tumor cells showed nuclear staining [16]. HER2 status was determined according to the American Society of Clinical Oncology (ASCO)/College of American Pathologists (CAP) guidelines [17]. A biopsy was considered HER2-positive with an IHC score of 3+, and HER2-negative with a 0 or 1+ score. In the case of an IHC 2+ score, HER2 dual brightfield in situ hybridization (INFORM dual BRISH, Ventana/Roche) was performed and also scored according to the ASCO/CAP guidelines [17]. Inferred molecular subtypes were defined in line with ESMO metastatic breast cancer guidelines [2]: ER-positive/HER2-negative, HER2-positive (ER-positive or negative), and triple-negative breast cancer (TNBC; ER-negative and HER2-negative). In addition, the metastasis was classified into histological subtypes, using the primary tumor histology as reference in some cases. Thereafter, tumor biopsy tissue was included in a tissue microarray (TMA) to allow additional histopathological analyses. From each formalin-fixed and paraffin-embedded tumor tissue block, three (or less if limited tissue was available) 0.6 mm cores containing representative tumor areas were obtained. Sections of 3 mm were serially cut and stained for Ki-67 (proliferation index) (30-9, Ventana/Roche) and cytokeratin 8/18 (CK8/18, B22.1, and B23.1, Ventana/Roche) on an automated immunostainer (Ventana Benchmark Ultra, Ventana/Roche). Whole-slide images were acquired using a Philips Ultra-Fast Scanner (Philips, Eindhoven, The Netherlands). Digital image analysis (DIA) to assess the Ki-67 proliferation index was performed using the Visiopharm Integrator System (VIS) version 2020.02.0.7219 (Visiopharm, Hørsholm, Denmark) and according to a validated assessment method described previously [18]. The total number of tumor cells on the TMA was determined for each patient. Biopsies with at least 100 tumor cells were included in the analyses. The Ki-67 proliferation index per patient was calculated by dividing the total number of Ki-67-positive cells by the total number of cells (at least ≥100 cells), thereby compensating for heterogeneous Ki-67 expression. A Ki-67 of ≥20% was considered high, and <20% low [19,20]. 

### 2.3. [^18^F]FDG-PET/CT Imaging

Whole-body (head to mid-femur) [^18^F]FDG-PET scans in the UMCG, Radboud UMC, and Erasmus MC were performed using a Biograph mCT 40 or 64-slice PET/CT camera (Siemens, Knoxville, TV, USA), and in the VUmc using an Ingenuity TF or Gemini TF PET/CT scanner (Philips). All cameras were European Association of Nuclear Medicine Research Limited (EANM/EARL)-accredited. Patients had to fast for at least 6h, and blood glucose levels had to be <11 mmol/L before tracer injection. Patients received an [^18^F]FDG bolus of 3 MBq/kg ± 10% intravenously 60 ± 5 min before PET/CT acquisition. PET scans with an acquisition time of 1–3 min per bed position were obtained in combination with low-dose CT for attenuation correction (sometimes in combination with a contrast-enhanced CT scan) and anatomical localization. Scan acquisition and reconstructions were performed following the recommendations of the EANM guidelines for oncologic [^18^F]FDG-PET imaging [21].

### 2.4. [^18^F]FDG-PET Analysis

The metastasis, from which a biopsy was taken, was selected as reference lesion for the analysis of [^18^F]FDG uptake. In the case that the location of the biopsied lesion was unclear from the medical reports, the exact location was verified by an experienced radiologist. The metastasis was considered qualitatively visible on [^18^F]FDG-PET if the uptake was visually higher than background activity. Quantitative analysis of the [^18^F]FDG-PET scans was performed according to the EANM guidelines for ^18^F-tracers on reconstructed images according to EARL [21]. Syngo.via VB20/30 imaging software (Siemens Healthineers, Knoxville, TN, USA) was used for quantification. Tracer uptake was quantified by trained readers (JB, BE, JG). A volume of interest (VOI) was manually drawn around the biopsied lesion based on a visible lesion on [^18^F]FDG-PET, in correlation with anatomical substrate on CT, and three types of SUVs (SUV_max_, SUV_peak_, and SUV_mean_) were calculated. A high correlation was found between the three SUV measurements (Appendix A); therefore, in the present analysis, only the SUV_max_ results are reported in line with the EANM guideline [21].

### 2.5. Statistical Analysis

We compared histopathological findings of the biopsied metastasis with [^18^F]FDG uptake of the corresponding lesion. We natural-log-transformed [^18^F]FDG uptake to obtain approximate normal distributions, yielding estimates of geometric means (SUV_max geom mean_) following back-transformation of the results. We related the [^18^F]FDG uptake level to hormone and HER2 receptor status, molecular subtypes, Ki-67 proliferation index, and histological subtype. This correlation was tested using the unpaired Student *t*-test (2 groups) or the One-Way ANOVA test with a Bonferroni post hoc test (>2 groups). The correlation between [^18^F]FDG uptake with Ki-67 and lesion size on CT as a continuous variable was assessed by Pearson’s correlation coefficient. To assess the difference in [^18^F]FDG-PET SUV_max_ within a patient, the median co-efficient of variability and median fold difference were calculated. To assess the discriminatory value of [^18^F]FDG uptake for histopathological characteristics, area under the receiver operating characteristic (ROC) curves (AUCs) for [^18^F]FDG uptake were assessed, including 95% confidence interval (CI). Fisher’s exact test and the Chi-square test were used for categorical data, namely, histological subtype versus HER2 status and Ki-67 proliferation index, respectively. Statistical significance was defined as a probability two-tail value of *p* ≤ 0.05. Statistical analyses were performed using IBM SPSS Statistics for Windows, version 23 (IBM Corp., Armonk, NY, USA), and R version 4.0.3 for Windows.

## 3. Results

### 3.1. Patients

Among 217 patients who signed informed consent forms, 15 patients did not meet inclusion criteria, 1 patient withdrew consent, and 1 patient refused a biopsy. In 188/200 patients the metastasis biopsy and [^18^F]FDG uptake of the metastasis were available for analysis (Figure 1). The characteristics of these 188 patients are shown in Table 1. Ninety-three biopsies were obtained from bone metastases. The median interval between [^18^F]FDG-PET and biopsy was one day with a range of –41 to +35 days. In total, 103 patients underwent the [^18^F]FDG-PET scan at least one day prior to the biopsy, 26 patients on the same day as the biopsy, and 59 patients at least one day after the biopsy. 

The order in which the diagnostics were performed did not influence the [^18^F]FDG uptake. The uptake in metastases biopsied 1–7 days before [^18^F]FDG-PET (*n* = 33/188) was similar to the uptake in metastases biopsied >7 days prior or after the [^18^F]FDG-PET (*n* = 129/188) with an estimated geometric mean SUV_max_ of 7.2 versus 7.4, respectively (*p* = 0.84). Patients who had a biopsy and [^18^F]FDG-PET on the same day (*n* = 26/188) were not included in this sub-analysis on the effect of the order of diagnostics. In total, 6 out of 188 biopsies were identified only on [^18^F]FDG-PET; these biopsies did not have different [^18^F]FDG uptake compared to the others (geometric mean SUV_max_ 9.3 versus 7.4; *p* = 0.37). Of 182 biopsied lesions containing tumor cells, 7 lesions were not detected on [^18^F]FDG-PET.

The accuracy/sensitivity of [^18^F]FDG-PET to predict tumor on CT was 96%, 175/182. The size of tumor lesions on CT (expressed as volume) (mean 19.8cc; range 0.1cc–588.9cc) was positively correlated with FDG uptake (R 0.27) (Appendix A).

The median co-efficient of variability to assess difference in [^18^F]FDG-PET SUV_max_ within a patient was 37.69 (min 1.04–max 112.7) and the median fold difference was 4.41 (min 1.00–max 18.26).

### 3.2. [^18^F]FDG Uptake and Non-Malignancy

In 6 out of 188 patients, no tumor cells were present in the biopsy: no representative tissue available (*n* = 2 bone, *n* = 1 lymph node, and *n* = 1 lung), no vital tumor cells in an inguinal lymph node (*n* = 1), and necrotizing granulomatous inflammation in an axillary lymph node (*n* = 1). The lesion from which a biopsy without vital tumor cells was obtained was not visible on [^18^F]FDG-PET, and was less than 10 mm on CT with an anatomical substrate. The remaining five lesions were all visible on [^18^F]FDG-PET.

### 3.3. [^18^F]FDG Uptake and Hormone and HER2 Receptor Status 

The estimated geometric mean SUV_max_ on [^18^F]FDG-PET was not significantly different between ER-positive (*n* = 133) and ER-negative (*n* = 49) tumors (7.1; 95% CI: 6.44–7.87 versus 8.4; 95% CI: 7.13–9.86; *p* = 0.087; weak discriminative power: AUC 0.58; 95% CI: 0.48–0.67; *p* = 0.12). Likewise, the estimated geometric mean [^18^F]FDG uptake was not significantly different between PR-positive (*n* = 112) and PR-negative (*n* = 70) tumors (7.1; 95% CI: 6.33–7.89 versus 8.1; 95% CI: 7.04–9.25; *p* = 0.13; weak discriminative power: AUC 0.57; 95% CI: 0.49–0.66; *p* = 0.10). The estimated geometric mean [^18^F]FDG uptake was higher in HER2-positive *(n* = 29) than in HER2-negative (*n* = 153) tumors (10.0; 95% CI: 8.19–12.25 versus 7.0; 95% CI: 6.41–7.71; *p* = 0.0064; moderate discriminative power: AUC 0.68; 95% CI: 0.57–0.79; *p* = 0.0019). Metastases were also classified as ER-positive/HER2-negative (*n* = 121), HER2-positive (*n* = 29), and TNBC (*n* = 32). A significant effect of inferred molecular subtype on [^18^F]FDG uptake was observed, with increasing geometric mean SUV_max_ in the following order: ER-positive/HER2-negative (6.8; 95% CI: 6.17–7.60), TNBC (7.6; 6.83–9.54), and HER2-positive (10.0; 8.19–12.25; *p* = 0.0058). A post hoc analysis showed that this effect was driven by the difference between the ER-positive/HER2-negative and HER2-positive molecular subtypes (*p =* 0.0045; Figure 2A).

### 3.4. [^18^F]FDG Uptake and Ki-67 Proliferation Index

Samples of 112 patients were available for Ki-67 proliferation index analysis. In 70 out of 182 tissues, no Ki-67 analysis could be performed, mainly due to insufficient tumor tissue to account for heterogeneity as described previously. In sixty-five samples, the Ki-67 proliferation index could be assessed on three different cores in the TMA. Forty-six tumors had a Ki-67 <20%, and sixty-six had Ki-67 ≥20%. A Ki-67 proliferation index ≥20% was observed more frequently in patients with HER2-positive (17 out of 21 patients; 81%) than HER2-negative tumors (49 out of 91 patients; 54%; *p* = 0.028). A weak positive correlation was found between the level of [^18^F]FDG uptake and the Ki-67 proliferation index (r = 0.34, *p* < 0.001; Figure 3). The estimated geometric mean SUV_max_ on [^18^F]FDG-PET was lower in patients with Ki-67 <20% (7.1; 95% CI: 6.22–8.03) than in patients with Ki-67 ≥20% (8.8; 95% CI: 7.79–10.04; *p* = 0.014; Figure 2B). [^18^F]FDG uptake showed a weak discriminative power for distinguishing metastases with high or low Ki-67 proliferation index (AUC 0.63; 95% CI: 0.52–0.73; *p* = 0.024).

### 3.5. [^18^F]FDG Uptake and Histological Subtype

The histological classifications were invasive carcinoma NST (146 tumors; 80%), invasive lobular carcinoma (31 tumors; 17%), and other types of carcinoma (five tumors; 3%; see Table 1). The estimated geometric mean SUV_max_ on [^18^F]FDG-PET was lower in invasive lobular carcinoma (5.8; 95% CI: 4.83–7.09) than invasive carcinoma NST (7.8; 95% CI: 7.06–8.56; *p* = 0.014; Figure 2C). This difference was not related to a difference in HER2 or Ki-67 status between the groups, because the percentage of HER2-negative invasive carcinoma NST was similar to the fraction of HER2-negative invasive lobular carcinoma (120/146; 82% versus 28/31; 90%, respectively; *p* = 0.44). Likewise, Ki-67 ≥20% was found in 59% of the invasive carcinoma NST cases (50/85) versus 61% of the invasive lobular carcinoma cases (14/23; *p* = 0.88). Despite the lower [^18^F]FDG uptake in invasive lobular carcinoma than in invasive carcinoma NST, the detection rate of biopsied metastases on [^18^F]FDG-PET was similar for invasive lobular carcinoma and invasive carcinoma NST, namely 97% versus 96%, respectively. Overall, in 175/182 patients (96%) the corresponding lesion showed [^18^F]FDG uptake above background on the PET scan (Appendix A). Figure 4 shows an example of a patient with an invasive lobular carcinoma with visible [^18^F]FDG uptake in all metastases. [^18^F]FDG uptake showed a weak discriminative power to distinguish invasive carcinoma NST from invasive lobular carcinoma (AUC 0.65; 95% CI: 0.55–0.75; *p* = 0.0094).

## 4. Discussion

In the present study, our results indicate that [^18^F]FDG-PET is a powerful tool to detect metastases in newly diagnosed non-rapidly progressive metastatic breast cancers of all subtypes. This study confirms the need for a biopsy to assess the histopathological characteristics of breast cancer; therefore, [^18^F]FDG-PET and a biopsy remain complementary in metastatic breast cancer staging.

This is the first prospective study, with fully standardized imaging and histopathological assessments, to relate known prognostic histopathological characteristics of breast cancer to [^18^F]FDG uptake in corresponding metastasis in a large homogenous cohort of patients with newly diagnosed non-rapidly progressive metastatic breast cancer. Of relevance to clinical practice and research are our findings that [^18^F]FDG-PET can be used in invasive lobular carcinoma, and that biopsies can be performed before or after the [^18^F]FDG-PET without affecting [^18^F]FDG uptake.

Mixed results about HER2 status and [^18^F]FDG uptake have been reported [5,7,8,22,23]. Differences in the studies may explain these results, for example, primary versus (first or later presentation of) metastatic disease, study size, and direct correlation between the biopsied site and imaging. In general, however, higher [^18^F]FDG uptake is found in HER2-positive compared to HER2-negative tumors, as described in a meta-analysis in primary breast cancers [24]. In the present study, HER2, but not ER or TNBC status, was associated with [^18^F]FDG uptake. Studies indicated a relation between molecular subtypes and [^18^F]FDG uptake in primary breast cancers [22,23,25]. But while our data indicated no relation between TNBC and [^18^F]FDG uptake, this relation was observed in retrospective studies [22,23,25]. In our study, the selection of non-rapidly progressive metastatic breast cancer may have played a role in these findings. The fact that we observed a high Ki-67 proliferation index more often in the HER2-positive than HER2-negative tumors, including TNBC, is in line with this. Similarly to our data, a positive relation between Ki-67 proliferation index and HER2, as well as between Ki-67 and [^18^F]FDG uptake, was previously described in (primary) breast cancers [26,27,28,29].

Another radiotracer that can be used as an imaging biomarker of proliferation is [^18^F]Fluorothymidine (FLT). A positive correlation has been found between [^18^F]FLT-PET uptake and Ki-67 expression in patients with breast cancer [30].

In our study, the histopathological characteristics of breast cancer could not be discerned with [^18^F]FDG-PET. Retrospective analyses of patients with primary breast cancer also showed that [^18^F]FDG uptake did not have enough discriminative power to identify ER- or HER2-positive tumors [29,31]. Therefore, although [^18^F]FDG-PET is clearly of value to determine the disease’s extent, i.e., the presence/location of (distant) disease, a biopsy remains necessary to assess the histopathological characteristics of breast cancer.

We found that the SUV_max_ value in invasive carcinoma NST was higher than in invasive lobular carcinoma, supporting the idea that histological subtype can affect [^18^F]FDG uptake [7]. Interestingly, the detection rate of metastases was similar for invasive carcinoma NST and invasive lobular carcinoma in our study. This supports the previous suggestion of a smaller study, with a 86% detection rate of [^18^F]FDG-avid metastases in invasive lobular carcinoma [32]. Therefore, [^18^F]FDG-PET can also be used for invasive lobular carcinoma in the metastatic setting. [^18^F]FES-PET may be of additional value to detect invasive lobular carcinoma, but this technique is less available than [^18^F]FDG-PET [33].

This study has inherent limitations. Only patients with non-rapidly progressive disease were included, which may have induced a selection bias. In the majority of patients, a single biopsy was performed and, therefore, intrapatient heterogeneity could not be observed by biopsies. However, heterogeneity could be assessed regarding the spread of [^18^F]FDG uptake. Furthermore, the maximum interval between [^18^F]FDG-PET and the biopsy was 6 weeks, and the metabolic rate of the tumor could have changed during this period. However, the median interval between [^18^F]FDG-PET and biopsy in this study was one day. Furthermore, a large metabolic difference is not expected due to the exclusion of patients with rapidly progressive disease. Another limitation is that the tumor Ki-67 proliferation index analysis could not be conducted in all patients. The strengths of this study include the large number of patients undergoing [^18^F]FDG-PET and a metastasis biopsy, allowing head-to-head comparison of tracer uptake and histopathological characteristics, including centralized PET quantification and centrally reviewed biopsies. The large sample size could be achieved by enrolling patients in four large Dutch university medical centers and an interinstitutional comparison of PET data was possible due to the multicenter harmonization for [^18^F]FDG-PET scans, guaranteed by the EARL accreditation. 

In conclusion, [^18^F]FDG-PET is a powerful tool to detect metastases, in newly diagnosed non-rapidly progressive metastatic breast cancer of all subtypes, including invasive lobular carcinoma. This prospective study confirms the need for a biopsy to assess the histopathological characteristics of breast cancer; therefore, [^18^F]FDG-PET and a biopsy remain complementary in metastatic breast cancer staging.

## Figures and Tables

**Figure 1 diagnostics-14-00416-f001:**
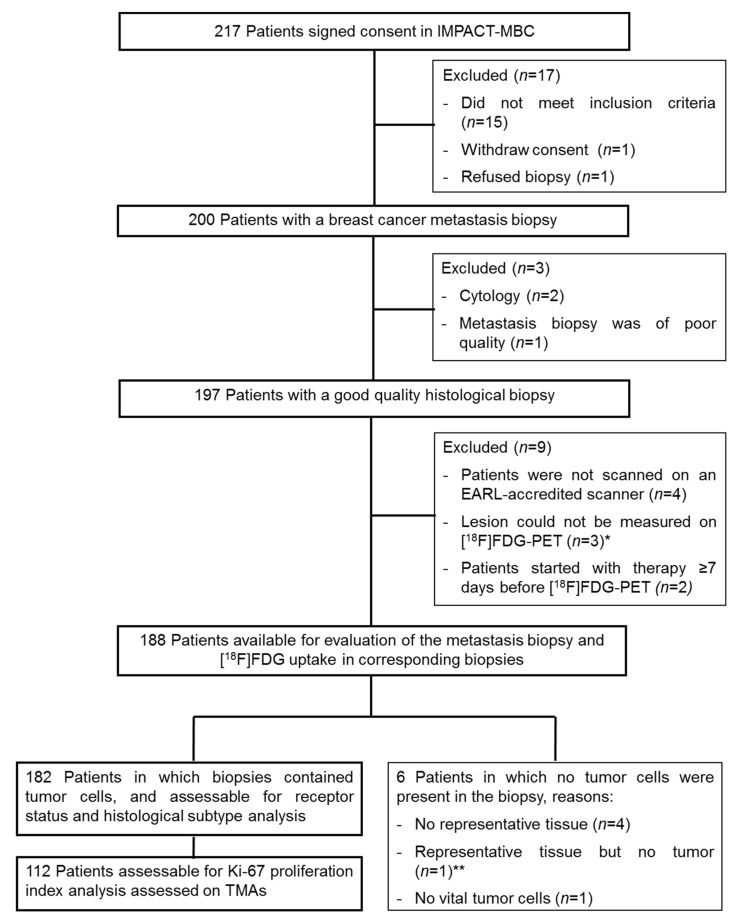
Consort diagram showing patients included in the IMPACT-MBC study and eligible for this analysis. * Skin, brain, and axillary lymph node (including presence of fibrosis) metastasis; ** necrotizing granulomatous inflammation in an axillary lymph node. Abbreviations: MBC, metastatic breast cancer; EARL, European Association of Nuclear Medicine Research Limited; TMA, tissue microarray.

**Figure 2 diagnostics-14-00416-f002:**
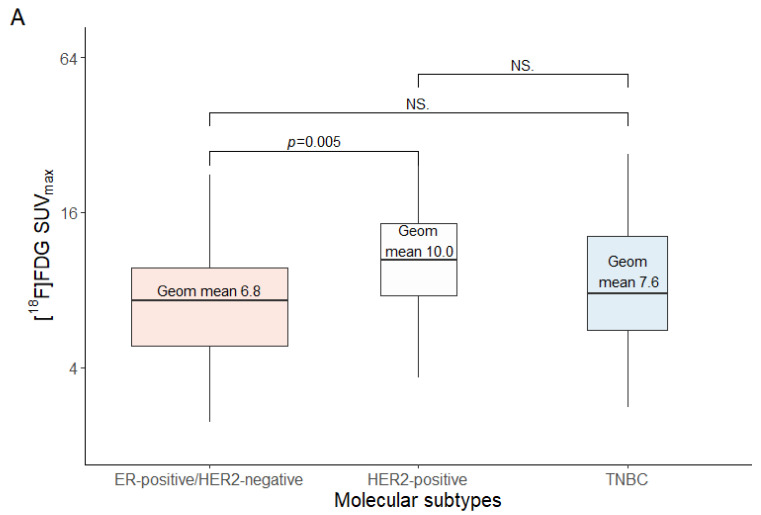
(**A**–**C**) [^18^F]FDG uptake (SUV_max_; plotted on a logarithmic scale) and (**A**) molecular subtypes, (**B**) Ki-67 proliferation index, and (**C**) histological subtype.

**Figure 3 diagnostics-14-00416-f003:**
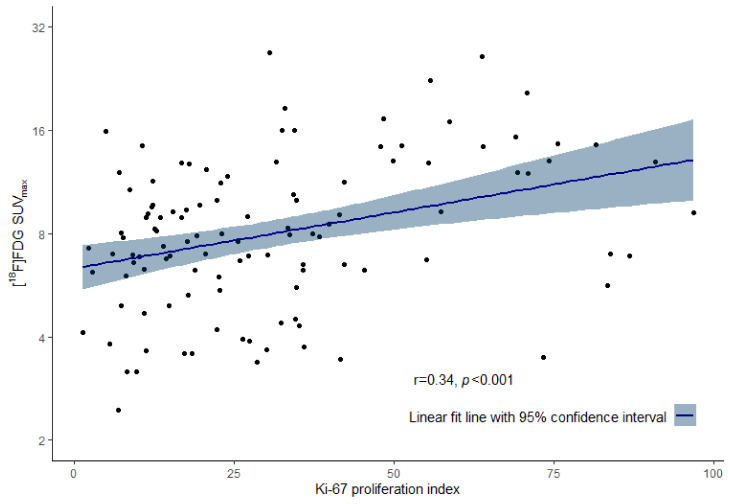
Correlation of [^18^F]FDG uptake (SUV_max_) and Ki-67 proliferation index (percentage).

**Figure 4 diagnostics-14-00416-f004:**
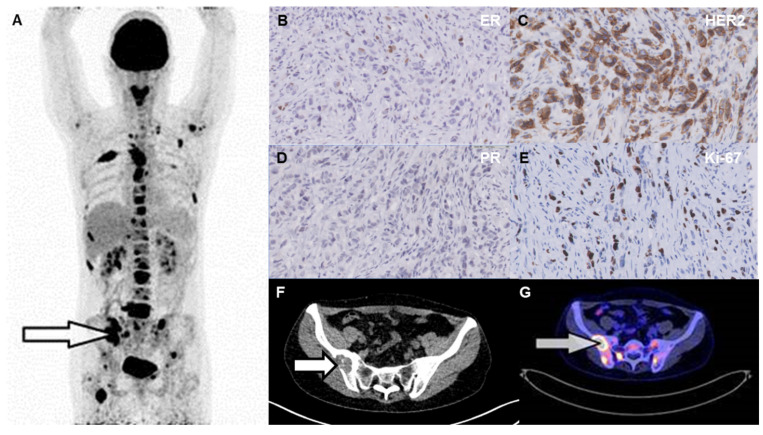
Example of a 46-year-old patient with an invasive lobular carcinoma with visible [^18^F]FDG uptake in all metastases. The whole-body [^18^F]FDG-PET showed metastases in lymph nodes and multiple bone lesions (image (**A**)). The arrows refer to the biopsied lesion of the right ilium on CT (image (**F**)), and intense [^18^F]FDG uptake (image (**A**,**G**)). This lesion was ER-positive (image (**B**)), PR-negative (image (**D**)), and HER2-positive (image (**C**)) with Ki-67 ≥20% (image (**E**)).

**Table 1 diagnostics-14-00416-t001:** Patient characteristics.

Patient Characteristic	Number (%) *n* = 188
Sex	
Female	186 (99)
Male	2 (1)
Age, years	
Mean (SD)	59 ± 11
Previous adjuvant therapies *	
None	52 (28)
Yes (including endocrine and/or chemotherapy and/or targeted)	136 (72)
Biopsy site of metastasis	
Bone	93 (50)
Lymph node	45 (24)
Liver	25 (13)
Lung	8 (4)
Skin	9 (5)
Other ^†^	8 (4)
Biopsy characteristics	
Biopsy containing tumor cells	182 (97)
ER status (*n* = 182)	
Positive	133 (73)
Negative	49 (27)
Median % [range] ^¶^	100 [0–100]
PR status (*n* = 182)	
Positive	112 (62)
Negative	70 (38)
Median % [range] ^§^	30 [0–100]
HER2 status (*n* = 182)	
Positive	29 (16)
Negative	153 (84)
IHC 0	68 (37)
IHC 1+	71 (39)
IHC 2+ - ISH+ *n* = 6 - ISH– *n* = 12 - ISH unknown *n* = 2	20 (11)
IHC 3+	22 (12)
IHC unknown - ISH+	1 (1)
Molecular subtypes (*n* = 182)	
ER-positive/HER2-negative	121 (66)
HER2-positive	29 (16)
- ER+ *n* = 12	
- ER- *n* = 17	
TNBC	32 (18)
Histology (*n* = 182)	
Invasive carcinoma NST	146 (80)
Invasive lobular carcinoma	31 (17)
Other ^ǁ^	5 (3)
Ki-67 proliferation index (*n* = 112)	
Ki-67 low (<20%)	46 (41)
Ki-67 high (≥20%)	66 (59)
Median % [range]	26 [1–97]

Abbreviations: SD, standard deviation; LN, lymph node; ER, estrogen receptor; PR, progesterone receptor; HER2, human epidermal growth factor receptor 2; TNBC, triple-negative breast cancer; IHC, immunohistochemistry; ISH, in situ hybridization; NST, no special type. * see Appendix A. ^†^
*n* = 1 adrenal gland; *n* = 2 breast; *n* = 3 thoracic soft tissue, *n* = 2 muscle soft tissue. ^¶^ The percentage of positive cells was not available (*n* = 1); ^§^ the percentage of positive cells was not available (*n* = 2). ^ǁ^
*n* = 1 tubulo-lobular carcinoma; *n* = 1 micropapillary carcinoma; *n* = 1 metaplastic carcinoma; *n* = 1 undifferentiated carcinoma; and *n* = 1 apocrine carcinoma.

## Data Availability

The dataset used for the current study is available from the corresponding author on reasonable request.

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
