# Peer review of "Correlation between Histopathological Prognostic Tumor Characteristics and [18F]FDG Uptake in Corresponding Metastases in Newly Diagnosed Metastatic Breast Cancer"

_diagnostics, 2024, doi:10.3390/diagnostics14040416_

Round 1

Reviewer 1 Report

Comments and Suggestions for Authors

This is essentially a negative study regarding the association between histopathological prognostic tumor characteristics and [18F]FDG uptake in corresponding metastases in newly diagnosed metastatic breast cancer. I think it is generally well done. I have two comments:

1) The authors could do a better job of explaining the physiology behind [18F]FDG uptake, regarding glucose metabolism, etc, which would help the reader to understand comments such as "Multiple histopathological characteristics of breast cancer, such as receptor status and Ki-67 proliferation index, have been described to affect the glucose metabolism of the tumor, resulting in differences in [18F]FDG standardized uptake values (SUVs)" (LINE 66-68)

2)The authors comment that "Therefore, although [18F]FDG-PET is clearly of value to determine the disease’s extent" (LINES 308-309). seems a bit overreaching to me, as there was no information on tumor size in the article. It would be acceptable to acknowledge [18F]FDG-PET is clearly of value to determine the presence of tumor, unless the author can better discuss how this technology relates to tumor extent. For example, does the geometric mean relate at all to tumor size? Better yet can the author present tumor size data and geometric mean?

Author Response

''Please see the attachment'' 

Reviewer 2 Report

Comments and Suggestions for Authors

The authors perform a correlation between FDG-PET and histopathology to detect metastases in breast cancer. While the study is interesting, there are some concerns. The efficacy of FDG-PET has been reported by multiple groups (PMID# 37019987, 28331746)

1) The authors should report sensitivity and accuracy of FDG-PET testing in their cohort. In line 88, it was mentioned CT was also performed on the patients included in their cohort. was a correlation between FDG-PET and CT performed in their study. If so, it should be reported. If not, it should be performed.

2) As this is a study performed between 2013 to 2018 do the authors have data on PERCIST and RECIST tumor evaluation criteria? If so, it would be good to provide that dat.

3) It is interesting that the authors saw a statistically significant difference in FDG-PET between Her2+ and Her2- patients. This is different from what is reported in #PMID 28331746. May be authors could comment on this difference.

Comments on the Quality of English Language

1) In general, the authors use informal English. This should be extensively modified across the manuscript. Just to give an example: line 58 it is written "....performed to evaluate the presence of tumor"... what authors meant was " biopsy is performed for detection of tumor..."; in line 59 the word 'status' is repeated back-to-back making it redundant; even the title uses the word "association" which is less scientific term, they should use the term 'correlation' (especially, figure 3 studies correlation).

To this end, I suggest getting the manuscript edited by scientific english writing experts.

2) The quality of figures in general are very poor. The font on axes is broken suggesting it is copy pasted to pdf from poorly exported quality original files.

Author Response

Thank you very much for taking the time to review our manuscript ‘Correlation between histopathological prognostic tumor characteristics and [18F]FDG uptake in corresponding metastases in newly diagnosed metastatic breast cancer.’ Please find the detailed responses below and the corresponding revisions in track changes in the re-submitted files.

Comment 1:

The authors perform a correlation between FDG-PET and histopathology to detect metastases in breast cancer. While the study is interesting, there are some concerns. The efficacy of FDG-PET has been reported by multiple groups (PMID# 37019987, 28331746)

1) The authors should report sensitivity and accuracy of FDG-PET testing in their cohort. In line 88, it was mentioned CT was also performed on the patients included in their cohort. was a correlation between FDG-PET and CT performed in their study. If so, it should be reported. If not, it should be performed.

Response 1: We would like to thank the reviewer for considering our study interesting. The references the reviewer mentions were both added on page 2, introduction, line 58 and 1 on page 13, discussion, line 312. We added sensitivity and accuracy of FDG-PET to predict tumor lesions on CT, on page 6, results, line 208-210. Also in response to reviewer 1, we added the correlation between FDG uptake and CT on page 6, results, line 210-212 and supplemental 2 (added).

Comment 2:

2) As this is a study performed between 2013 to 2018 do the authors have data on PERCIST and RECIST tumor evaluation criteria? If so, it would be good to provide that dat.

Response 2: Thank you for pointing this out. RECIST 1.1 was used to define target lesions, as now more clearly described in the methods section, page 3, methods, line 101-102. The present paper is focused on baseline data, as the outcome/response data are currently still being analyzed. This is now also further clarified on page 3, methods, line 104-105.

Comment 3:

3) It is interesting that the authors saw a statistically significant difference in FDG-PET between Her2+ and Her2- patients. This is different from what is reported in #PMID 28331746. May be authors could comment on this difference.

Response 3: Thank you for pointing this out. This reference was added and the possible causes for differences with our findings were further described in the discussion section, page 13, discussion, line 312-315.

Comment 4. Response to Comments on the Quality of English Language

Point 1: In general, the authors use informal English. This should be extensively modified across the manuscript. Just to give an example: line 58 it is written "....performed to evaluate the presence of tumor"... what authors meant was " biopsy is performed for detection of tumor..."; in line 59 the word 'status' is repeated back-to-back making it redundant; even the title uses the word "association" which is less scientific term, they should use the term 'correlation' (especially, figure 3 studies correlation). To this end, I suggest getting the manuscript edited by scientific english writing experts.

Response 4: Thank you for pointing this out. We adapted the text and language as suggested by the reviewer, and performed a further external review for language quality.

Comment 5. Response to figures

Comment 1: The quality of figures in general are very poor. The font on axes is broken suggesting it is copy pasted to pdf from poorly exported quality original files.

Response 5: Thank you for pointing this out. To ensure good quality files we added the original Figures.

Round 2

Reviewer 2 Report

Comments and Suggestions for Authors

No further comments

Author Response

Thank you for your review